# MOTI$\mathcal{VE}$: A Drug-Target Interaction Graph For Inductive Link Prediction

**John Arevalo**[*]   **Ellen Su**[*]   **Anne E. Carpenter**   **Shantanu Singh**

Broad Institute of MIT and Harvard

{jarevalo, suellen, anne, shantanu}@broadinstitute.org

## Abstract

Drug-target interaction (DTI) prediction is crucial for identifying new therapeutics and detecting mechanisms of action. While structure-based methods accurately model physical interactions between a drug and its protein target, cell-based assays such as Cell Painting can better capture complex DTI interactions. This paper introduces MOTI$\mathcal{VE}$, a **M**orphological c**O**mpound **T**arget **I**nteraction **Graph** dataset comprising Cell Painting features for $11,000$ genes and $3,600$ compounds, along with their relationships extracted from seven publicly available databases. We provide random, cold-source (new drugs), and cold-target (new genes) data splits to enable rigorous evaluation under realistic use cases. Our benchmark results show that graph neural networks that use Cell Painting features consistently outperform those that learn from graph structure alone, feature-based models, and topological heuristics. MOTI$\mathcal{VE}$ accelerates both graph ML research and drug discovery by promoting the development of more reliable DTI prediction models. MOTI$\mathcal{VE}$ resources are available at `https://github.com/carpenter-singh-lab/motive`.

## 1 Introduction

High-quality graph benchmarking datasets propel graph machine learning (ML) research. Providing a diversity of domains, tasks, and evaluation methods, they allow for rigorous and extensive explorations of structured learning methods. Still, gaps remain in the areas of scalability, network sparsity, and generalizability under realistic data splits [1]. These challenges are particularly relevant in the biological domain. The representation of the rich heterogeneity between entities—compounds, genes, proteins, diseases, phenotypes, side effects, and more—is a nontrivial task due to their varied units and terminology and highly complex relational structure. This makes biological data an apt, challenging, and bettering application for graph ML.

Next, effectively predicting drug-target interactions (DTIs), the relationships between chemical compounds and their protein targets, remains a pressing research area due to its relevance to drug discovery, drug repurposing, understanding side effects, and virtual screening. The DTI task is challenging due to the shortage of clean perturbational data and nonspecificity of these interactions. Even as structure-based methods such as AlphaFold3 are increasingly accomplished at making DTI predictions, they are mainly based on molecular characteristics [2, 3]. Experimental data uniquely captures complex biological interactions; the morphological profiles, feature vectors that capture a cell's appearance, from the Cell Painting (CP) assay have been shown to model the mechanism of action, toxicity, and additional properties of compounds [4, 5].

To address the challenges of graph ML, biological data representation, and DTI, we introduce a publicly available dataset, MOTI$\mathcal{VE}$, which enhances a graph of compound and gene relations with features from the JUMP Cell Painting dataset [6]. As there is currently no compound-gene graph

---

[*]Equal contribution

dataset containing empirical node features, MOTI$\mathcal{VE}$ will be extremely useful for inductive graph learning [5] (generalizing to newly connected nodes), cold start recommendations [7] (generalizing to isolated nodes), and zero-shot scenarios [8] (generalizing to isolated node pairs). In many domains, making predictions for least-known entities are the most useful real-world applications [9]. Thus, we accompany MOTI$\mathcal{VE}$ with a rigorous framework of data splitting, loading, and evaluation. This work advances both DTI by incorporating a new modality of information and the strength of graph ML, as it rises to the challenge of knowledge generalization for inductive link prediction.

## 2    Related work

Although many graph-based datasets exist, Hu et al. [1] notes that there is a trade-off between scale and availability of node features. The Open Graph Benchmark Library (OGBL) thus contributed `ogbl-ppa`, `ogbl-collab`, and `ogbl-citation2`, all large-scale, feature-based link prediction datasets. The node features in each of these datasets are 58- or 128-dimensional and are a one-hot representation of the protein type in `ogbl-ppa` or `Word2Vec`-based representations of an author's publications or of a paper's contents in `ogbl-collab` and `ogbl-citation2` respectively. The benchmarking results showed a continued reliance on model learning from previous connections rather than features, as evidenced by the high performance of `Node2Vec` embeddings in OGBL tasks, and indicated a need for richer features. The authors also reported that the graph neural network (GNN) models underperformed in the link prediction task when using mini-batch training rather than whole batch and called for improvement in this area for future scalability when learning on large datasets. In addition, the latter two datasets split their data by time metadata associated with each edge, and did not explore cold start splits. Evidently, the field still requires graph datasets that 1) are large-scale, 2) include information-rich features, 3) are accompanied by graph-based data splits (not metadata-based), and 4) are flexibly trained with mini-batch sampling. We prioritized all four goals during the curation of MOTI$\mathcal{VE}$ and in our experimental design.

As Knowledge Graphs (KGs) have emerged as powerful tools for representing and learning from network-based data [10], they have often been applied to biological and chemical tasks such as drug discovery, structure prediction, and DTI [11–13]. Recent surveys on graph-based methods for DTI prediction [14, 15] show few efforts represent drugs and genes with external features. Two related approaches are Schwarz et al. [16], which used gene expression as input to the last layer in a late-fusion manner, and Balamuralidhar et al. [17], which used topological features as node representations. While Balamuralidhar et al. [17] does enable inductive predictions on newly connected nodes, it fails to make inferences on completely isolated nodes in cold start scenarios. To our knowledge, no graph-based dataset exists where the features of compounds and gene nodes are represented by their morphological profiles.

Meanwhile, in biological image analysis, fluorescence labeling of cells now allows for the visualization of cell morphology, internal structures, and processes at unprecedented spatial and temporal resolution. Imaging precisely captures the changes to a cell after it has undergone a chemical or genetic perturbation and sheds light on relationships such as DTI, functions, and mechanisms. Additionally, the recent publication of the JUMP Cell Painting dataset [6] now provides $136,000$ chemical and genetic perturbational profiles for the DTI task. In recent applications of CP to DTI, Rohban et al. [18] matched compounds to a small set of genes based on morphological feature vector similarities, and we will extend this work by using machine learning to capture non-linear compound-gene relationships for a much larger gene set. Next, Herman et al. [4] developed a deep learning model to predict toxicity assays from chemical structures and morphological profiles but did not exploit the connectivity network of compounds and gene interactions. Our method builds on this approach by incorporating the morphological profiles of both compounds and genes in a graph and using the message-passing framework to leverage such network connectivity. Recently, Iyer et al. [19] formulated the DTI task as a binary classification of gene-compound pairs under different data splits and developed a transformer-based learning approach to predict drug targets from Cell Painting profiles. Although this approach is similar to our proposed setup, their dataset contains fewer nodes, 302 compounds and 160 genes, and does not include gene-gene and compound-compound interactions.

The graph ML community needs large-scale datasets with rich, empirical node features, and well-defined data splitting, loading, and evaluation procedures that advance scalability (training in batches) and generalizability (inductive link prediction). The drug discovery community is motivated by

reliable DTI predictions, especially for newly discovered or assayed compounds and genes with no known relationships. The biological image analysis community is likewise curious about the reach, applicability, and predictive power of the morphological profiles of perturbed cells. To address these needs across multiple domains, we contribute the MOTI$\mathcal{VE}$ dataset.

## 3    The MOTI$\mathcal{VE}$ dataset

The MOTI$\mathcal{VE}$ dataset leverages drug-target knowledge graphs and image-based profiles of chemically or genetically perturbed cells. It comprises $11,509$ genes, $3,632$ compounds, and $303,156$ compound-compound, compound-gene, and gene-gene interactions collected from seven publicly available databases.

### 3.1    Morphological profiles extraction

Every gene (each of which produces a particular protein) and compound is represented with its image-based profile from the JUMP CP dataset [6]. JUMP CP images capture a cell's morphology after being perturbed by a chemical compound or a genetic edit. Each type of genetic perturbation, CRISPR knockouts or ORF over-expressions, involves a different set of genes. We created and published a version of MOTI$\mathcal{VE}$ for each gene set but chose to use the ORF genes for the analysis presented in this paper due to slightly stronger downstream performance. The morphological profiles were extracted from each image in the JUMP CP dataset using CellProfiler [20], a software that segments individual cells in the image and measures thousands of features for each cell. Then, the data was prepared according to the protocols in Arevalo et al. [21] and Chandrasekaran et al. [6], which extensively optimize separate pipelines for compound and gene perturbations.

For compound perturbations, we first filtered out features with low variance, then applied median absolute deviation normalization, a rank-inverse normal transformation (INT), and Harmony [22] to reduce the batch effects. We then selected final features based on correlation analysis. For genetic perturbations, we subtracted the mean vector per well from each feature vector to account for well position effects, replaced the INT transformation with an outlier removal step, and aggregated the replicates (usually five) of each perturbation using median profiles to make the representations robust to low-quality images. In our experience, these comprised less than $5\%$ of the data and were uncorrelated across replicates on different plates. After the correction and preprocessing steps, each compound and gene is represented by a 737-dimensional and 722-dimensional vector, respectively. The processing pipelines for the morphological profiles are available at `https://github.com/broadinstitute/jump-profiling-recipe/tree/v0.1.0`.

### 3.2    Annotation collection

| Database | compound gene | compound compound | gene gene | compound identifier |
|---|---|---|---|---|
| BioKG [23] | ✓ | ✓ | ✓ | DrugBank |
| DGIdb [24] | ✓ | | | ChEMBL |
| DRHub [25] | ✓ | | | PubChem |
| Hetionet [12] | ✓ | ✓ | ✓ | DrugBank |
| OpenBioLink [26] | ✓ | | ✓ | PubChem |
| PharMeBINet [27] | ✓ | ✓ | ✓ | DrugBank |
| PrimeKG [11] | ✓ | ✓ | ✓ | DrugBank |

Table 1: Databases integrated in MOTI$\mathcal{VE}$. We extracted all compound-gene, compound-compound and gene-gene annotations. Compound IDs were mapped to the InChIKey representations, and gene IDs were mapped to the NCBI gene symbols.

We aggregated the compound-compound, compound-gene and gene-gene annotations in MOTI$\mathcal{VE}$ from seven publicly available databases listed in Table 1. Our choice followed the comprehensive review of Bonner et al. [28], which categorizes relevant biomedical datasets and KGs based on the

entities they contain and how well they fit to specific tasks. We unified the compound IDs by mapping them to their core molecular skeleton identifiers (InChIKey) using the MyChem [29] and UniChem [30] databases and unified the gene IDs using their NCBI gene symbols [2]. We then combined all of the pairwise interactions we collected across databases with the JUMP CP features, keeping only the pairs where both entities had existing features and using only the pairs for which we had interactions.

## 3.3 Graph construction

Since DTI is defined by identifying the protein targets for a given drug, we assigned nodes associated with compounds as source nodes and nodes associated with genes as target nodes. We refer to compound-gene, compound-compound, and gene-gene as source-target, source-source, and target-target interactions respectively. We defined each source or target as a node and each interaction as an undirected edge. The representation of each node was set as its processed morphological feature vector as described in Section 3.2.

We defined our graph $G = (\mathcal{V}, \mathcal{E})$ as follows. $\mathcal{V} = \mathcal{S} \cup \mathcal{T}$ is the union of the sets of sources and targets in our dataset. Every $s \in \mathcal{S}$ and $t \in \mathcal{T}$ is represented by a feature vector $x_s \in \mathbb{R}^n$ and $x_t \in \mathbb{R}^m$, respectively. The edge set $\mathcal{E} = SS \cup ST \cup TT$ is the union of our source-source, source-target, and target-target edges, where each edge is defined as a pair of node indices: $(s_u, s_v) \in SS$, $(s_u, t_v) \in ST$, and $(t_u, t_v) \in TT$. We defined four different graph structures to assess the information added by each of our edge types: a *bipartite* graph which only includes $ST$ edges, a source-expanded (*s_expanded*) graph which includes $ST$ and $SS$ edges, a target-expanded (*t_expanded*) graph which includes $ST$ and $TT$ edges, and a source and target expanded (*st_expanded*) graph which includes all three edge types (statistics in Table 2). Note that, unless otherwise stated, our evaluated models received the *st_expanded* graph as it maximized the available information in MOTI$\mathcal{VE}$.

| Graph Type | # Nodes | # ST Edges | # Other Edges | Avg. Node Degree | Med. Node Degree |
|---|---|---|---|---|---|
| bipartite | S: 2,961 
 T: 4,505 | 24,798 | SS: 0 
 TT: 0 | S: 8.4 
 T: 5.5 | S: 3.0 
 T: 3.0 |
| s_expanded | S: 3,632 
 T: 4,505 | 24,798 | SS: 75,330 
 TT: 0 | S: 48.3 
 T: 5.5 | S: 19.5 
 T: 3.0 |
| t_expanded | S: 2,961 
 T: 11,509 | 24,798 | SS: 0 
 TT: 203,028 | S: 8.4 
 T: 37.4 | S: 3.0 
 T: 25.0 |
| st_expanded | S: 3,632 
 T: 11,509 | 24,798 | SS: 75,330 
 TT: 203,028 | S: 48.3 
 T: 37.4 | S: 19.5 
 T: 25.0 |

Table 2: Statistics for the four graph structures.

## 3.4 Data splitting

We developed two different heuristics to split our graph into training, validation, and test sets based on the random split and cold start split originally defined for recommendation systems [7].

In the random split scenario, we used a 70/10/20 ratio to randomly select and split every $ST$ edge into train, validation, and test sets and included the full set of $SS$ and $TT$ edges in training (Figure 1). In this case, the link prediction task is transductive, as the nodes are fixed from the start of training and the model will predict edges on entities it has learned on. Negative edges are sampled for each batch of data by randomly selecting source-target pairs that are not in the $ST$ edges.

The cold start split allows for inductive link prediction, as it involves predicting edges where at least one node was not present during training. We applied the cold start split to either the source or target nodes, denoted by cold-source split or cold-target split. In cold-source split (second row of Figure 1), every source node in our graph is randomly labeled as either train, validation, or test in a 70/10/20 ratio. All $ST$ edges are subsequently labeled in accordance with the label of its source node. Next, all $SS$ edges are labeled by their most conservative labels: any edge with at least one test source is

---

[2] https://www.ncbi.nlm.nih.gov/gene/

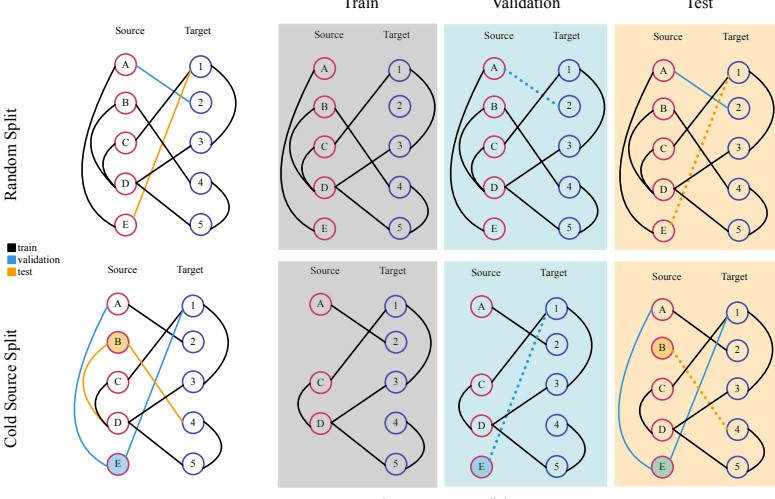

Figure 1: Schematic of the random split (top row) and cold-source split (bottom row). The left-most graph illustrates the actual partitioning of the edges, and the three graphs to the right show which nodes and edges are visible to the GNN models during training, validation, and testing. The number of edges in each partition is not representative of our true 70/10/20 ratios. Cold-target split is symmetrical to cold-source split. The model aggregates neighbor features via the message-passing edges (solid lines) and makes predictions on the supervision edges (dotted lines).

labeled test, any edge with no test sources but at least one validation source is labeled validation, and any edge with two train sources is labeled train. All $TT$ edges are included in training. This data split prevents data leakage of any kind and enforces that at validation or test time, the model has never seen any source node in the $ST$ edges. Additionally, negative edges are sampled for each batch of data by our custom sampler class, built for inductive learning, which only considers the sampled source nodes and all of target nodes, in the $ST$ edges. Details about our negative sampling algorithm can be found in Section 3.5.

Cold-target split is constructed in a symmetrical manner to cold-source split. The cold splits are useful for considering how the model will generalize to unseen and isolated sources and targets. When the node embeddings represent real-world features, models trained under this stringency will be especially useful in predicting novel yet relevant relationships from empirical data.

## 3.5 Negative sampling algorithm

Scalable gradient-based methods process data in randomly sampled batches, or subgraphs in the graph domain. Link prediction approaches commonly generate negative edges by dynamically sampling disconnected nodes. To prevent data leakage, the sampled negative edges during training must not include cold instances from validation or testing. However, we observed that the negative sampling strategy in PyG [31], a widely used GNN framework, lacks the granularity to control the sampled population. To address this limitation, we developed a custom negative sampling algorithm that guarantees test node isolation and enables proper evaluation of link prediction models.

We used a negative sampling ratio ($r$) of 1:1 during training and 1:10 during testing. For random split, our algorithm samples the negative edges between all unique source and target nodes within the $ST$ edges in the batch. For cold-source split, the head of the negative edge is sampled from the unique sources in the supervision $ST$ edges in the batch, and the tail of the negative edge is sampled from $ST$ edges in the batch. Algorithm 1 details the cold-source split negative sampling procedure. The process is symmetrical for leave-out-target.

**Algorithm 1:** Negative sampling for cold-source split

**Input:** $ST = \{(s_u, t_v) \mid \exists$ an edge between source $u$ and target $v\}$: all ground truth $ST$ edges

$\qquad B_P \subseteq ST$: positive supervision edges in batch

$\qquad r \in \mathbb{Z}_{>0}$: negative sampling ratio

$\qquad f(A, k)$: sample $k$ elements from set $A$

$\qquad n$: number of sampling tries

**Output:** $B'$: positive and negative supervision edges in batch

$c \leftarrow |B_P| \qquad\qquad\qquad\qquad\qquad \triangleright$ # positives;

$S_B \leftarrow \{s \mid \exists(s,t) \in B_P\}$;

$T_{ST} \leftarrow \{t \mid \exists(s,t) \in ST\}$;

**for** $i := 0$ **to** $n$ **do**

$\quad B_N \leftarrow f\left(\{(s,t) \mid s \in S_B \land t \in T_{ST}\}, 2cr\right) \qquad \triangleright$ Sampled negative edges;

$\quad B_N \leftarrow B_N \setminus ST$ ;

$\quad$**if** $|B_N| \geq cr$ **then**

$\quad\quad B'_N \leftarrow f(B_N, cr)$;

$\quad\quad B' \leftarrow B_P \cup B'_N$;

$\quad\quad$**return** $B'$;

$\quad$**end**

**end**

## 3.6 Models

We experimented with three different types of GNN convolutional layers in our model, each with a unique learning algorithm. The algorithms differ in how they incorporate the features of neighbor nodes into a node's learned representation. GraphSAGE (Graph SAmple and aggreGatE) [5] achieves inductive representation learning on large graphs by sampling from neighbor node representations and applying an aggregation function (e.g. a weighted average) to compute each node's hidden embedding. GIN (Graph Isomorphism Network) [32] adds a Multilayer Perceptron (MLP) to represent the composition of neighbor node features and achieves discriminative power equal to the Weisfeiler Lehman graph isomorphism test. Finally, GATv2 (Graph Attention Network v2) [33] implements dynamic graph attention which weights neighbor representations according to each query of interest. We constructed separate models using each of these GNN algorithms (GraphSAGE$_{CP}$, GIN$_{CP}$, and GATv2$_{CP}$) as the convolution layer, and we used the JUMP Cell Painting features as the nodal representations for each. We chose to only initialize the node features using their image-based features (as opposed to chemical structure-based or protein structure-based) so that compounds and genes would share a modality of representation. This ensures similar statistical properties between node types and makes it easier to discover cross-modal relationships [34].

All three GNN models share the same architecture. First, the input embeddings of every source and target node are initialized and fixed as their respective feature vectors $x_s$ and $x_t$. Two linear layers then transform the source and target node embeddings into the same embedding space. These transformed embeddings pass through two GNN layers (chosen from one of the three graph convolutional algorithms above), separated by a leaky ReLU activation function for nonlinearity. Broadly, each GNN layer combines the feature vector of the node of interest with some aggregation of the feature vectors from the neighbors of the node of interest. The aggregation function changes according to the three GNN algorithms. Isolated nodes rely solely on CP features due to the lack of neighboring signals. The two GNN layers indicate that the feature vectors of neighbor nodes within a radius of 2 will be used to compute the hidden embedding for each node. An additional skip connection feeds the output of the first GNN layer (an aggregation of the feature vectors in the neighborhood of radius 1) into the final embedding, prioritizing shorter distance neighbors. Finally, a classifier head outputs the sigmoid-transformed dot product of the embeddings for each source and target node pair in the supervision edge set. Algorithm 2 in Appendix A further describes the forward pass of our GNNs to make the link predictions.

We benchmarked our model with a featureless graph-based model that randomly initializes the source and target input embeddings. We opted to use the simple GraphSAGE convolutional layer in our model because all of the node features are random vectors in this case. This model, GraphSAGE$_{embs}$,

shares the same architecture as the other GNNs. Importantly, GraphSAGE$_{embs}$ does not have informative feature vectors for each source and target node at the start of training. We also applied one baseline heuristic for topology-based link predictions, and two baseline models for feature-based link predictions. For the former, we used the shortest path between source and target nodes as a proxy for their similarity to evaluate the predictive power of graph structure alone. For the latter, we implemented a Bilinear model and an MLP, which both rely solely on node features to predict links. The Bilinear model learns a mapping $y = x^T W z$ to transform two sets of feature vectors into the same embedding space, then computes the similarity between the vectors. The MLP learns hidden embeddings of the feature vectors via two fully connected layers and ReLU activation functions for nonlinearity, then uses a Bilinear head to make link predictions. These latter models ignore the known relational information between entities when predicting links. Across the seven models, we could readily evaluate the performance enhancements coming from adding node features or graph structure and determine the most useful learning approach for each of our scenarios. See Appendix B for experimental details and evaluation metric choices.

## 4 Results

### 4.1 DTI prediction improves with CP features

| Input | Model | F1 | Hits@500 | Precision@500 |
|---|---|---|---|---|
| | GIN$_{CP}$ | **0.5238**± 0.040 | **0.4552** ±0.017 | **0.9920** ±0.005 |
| Graph+CP | GATv2$_{CP}$ | 0.4169±0.007 | 0.2219±0.018 | 0.8452±0.045 |
| | GraphSAGE$_{CP}$ | 0.3836±0.029 | 0.2637±0.023 | 0.9056±0.012 |
| CP | MLP | 0.3829±0.008 | 0.2545±0.011 | 0.8456±0.015 |
| | Bilinear | 0.1703±0.003 | 0.0213±0.001 | 0.1812±0.008 |
| Graph | GraphSAGE$_{embs}$ | 0.3456±0.006 | 0.2254±0.013 | 0.8476±0.029 |
| | Shortest Path | — | 0.0025 | 0.0 |

Table 3: Test metrics (all with maximum value=1), averaged over 5 runs, for all models in the random split scenario. GNNs initialized with Cell Painting data are indicated with a $CP$ subscript. The shortest path heuristic is fixed across runs, and thus does not have standard deviation values. In addition, only rank based metrics were computed for shortest path since there is no classification threshold. The metric parameter k=500 corresponds to the top 1% of the test edges.

First, we evaluated all of our models on transductive link predictions (random split). We used the shortest path heuristic, a non-learning baseline, to predict links based on graph topology (last row of Table 3). We scored each positive and negative $ST$ edge in our test set by the shortest path length between the two nodes and computed rank based metrics based on these scores. This heuristic fails to effectively predict links on MOTI$\mathcal{VE}$, proving the task nontrivial and indicating a need for learning-based methods. The low prediction scores from the Bilinear and MLP models (rows 4 and 5 of Table 3) also indicate that predicting edges purely based on the similarity of source and target node features is inadequate. Still, these models significantly outperform the non-learning baseline, signaling the information contained in the node features.

Next, we see a large score increase from the feature-based benchmarks to the GNN models, which make use of the graph structure and CP node features. The predictions made by the learned representations of source and target nodes from topology alone (GraphSAGE$_{embs}$) achieve a Precision@500 score of 84.76%, which highlights the richness of the relational information between nodes. Finally, the GIN$_{CP}$ model achieves the best performance across metrics, obtaining a Hits@500 score of 45.54% and a Precision@500 score of 99.20%. This result supports our hypothesis and demonstrates that adding CP node features to graph structure benefits transductive link prediction.

### 4.2 Inductive link prediction benefits from graph structure and CP features

We evaluated the models on inductive link prediction tasks using the cold data splits, which require informative node features as the left out nodes are completely isolated during evaluation [5]. In this scenario, GraphSAGE$_{embs}$ is not applicable as it learns representations for nodes using projection

| Split | Model | F1 | Hits@500 | Precision@500 |
|---|---|---|---|---|
| Source | $\text{GIN}_{CP}$ | **0.2827**±0.018 | **0.1078**±0.008 | **0.5872**±0.028 |
| | $\text{GraphSAGE}_{CP}$ | 0.2283±0.014 | 0.0439±0.013 | 0.3120±0.079 |
| | $\text{GATv2}_{CP}$ | 0.2433±0.009 | 0.0330±0.002 | 0.2476±0.004 |
| | MLP | 0.1876±0.006 | 0.0269±0.009 | 0.2168±0.063 |
| | Bilinear | 0.1593±0.004 | 0.0145±0.003 | 0.1256±0.029 |
| Target | $\text{GIN}_{CP}$ | **0.3916**±0.016 | **0.2744**±0.020 | **0.9624**±0.034 |
| | $\text{GraphSAGE}_{CP}$ | 0.3494±0.012 | 0.1988±0.013 | 0.7056±0.031 |
| | $\text{GATv2}_{CP}$ | 0.2805±0.007 | 0.1966±0.019 | 0.9540±0.043 |
| | MLP | 0.3408±0.014 | 0.1837±0.010 | 0.6552±0.025 |
| | Bilinear | 0.1396±0.001 | 0.0122±0.001 | 0.1024±0.004 |

Table 4: Test metrics (all with maximum value=1), averaged over 5 runs, for all feature-based models. The top five rows show the cold-source split results and the bottom five rows show the cold-target split results.

layers and is thus limited to making predictions for nodes that it has already seen during training (i.e. transductive link predictions). The shortest path heuristic is also not applicable, as it is unable to handle isolated nodes unreachable by any path.

During inductive link prediction, all of the models only have access to the feature vectors of left-out nodes, as they are isolated and without edge connections. For both cold-source and cold-target split, $\text{GIN}_{CP}$ greatly outperformed all other models. The improved predictions by $\text{GIN}_{CP}$ can be attributed to it having learned improved source and target node embeddings by leveraging both the node features and the relational information present in the graph structure during training. Except for the F1 score of the GATv2 model in the cold-target split, all of the GNN models outperform the feature-based models. This supports our hypothesis that knowing the relationships between sources and targets improves DTI prediction.

Inductive link prediction is useful in real world applications as the model is learning about and making predictions on entities that we previously knew very little about. The cold-source split is especially relevant for drug discovery, as it simulates the scenario where all genes are known and the DTI model is tasked with identifying new compounds associated with certain disease-related genes. Furthermore, empirical features, such as morphological profiles, prove to be valuable node representations. If we collect experimental data for a newly discovered compound or unexplored gene, $\text{GIN}_{CP}$ and similar models would be able to leverage known relationships between other compounds and genes to make better inductive link predictions for the new entity. This underscores the value of integrating assay-based data into graph-based models to enhance their predictive capabilities.

### 4.3 Ablation studies with graph structure

We investigated the contribution of each edge type (source-target, source-source, and target-target) by comparing the performances of the GraphSAGE models using the four graph structures (Figure 2). While this analysis can apply to any of the GNN models, we chose GraphSAGE such that we could compare the feature-based and embeddings-based cases. We see that $\text{GraphSAGE}_{CP}$ outperforms $\text{GraphSAGE}_{embs}$ for almost every graph structure and metric. More analyses should be done to investigate how the difference in performance fluctuates with the addition of edge types in order to understand how the usefulness of the node features fluctuates with graph sparsity. From these initial results, it appears that as the graph becomes denser with the addition of more edge types, the structural information begins to dominate, reducing the relative contribution of the node features. These findings highlight the complementary nature of node features and graph structure in the link prediction task.

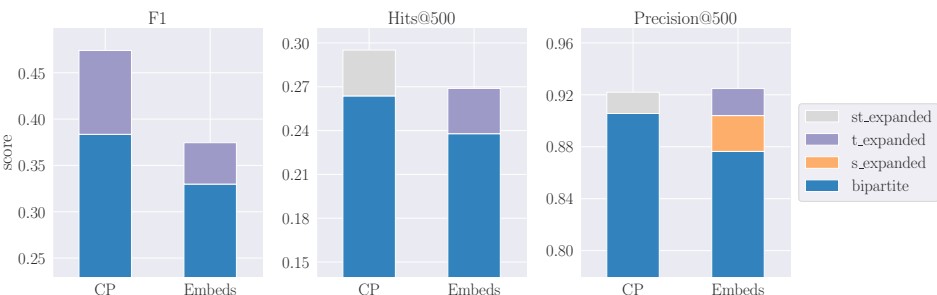

Figure 2: Test metrics for GraphSAGE$_{CP}$ and GraphSAGE$_{embs}$ for all four graph structures, with random data splits, and averaged over 5 runs. The colored stratifications of each bar show the decreasing performances of the models as edge types are removed from the graph. Note that the bars of each color are overlaid onto each other in the order specified in the legend, such that a structure color will only appear if it obtained worse performance than the previous structure.

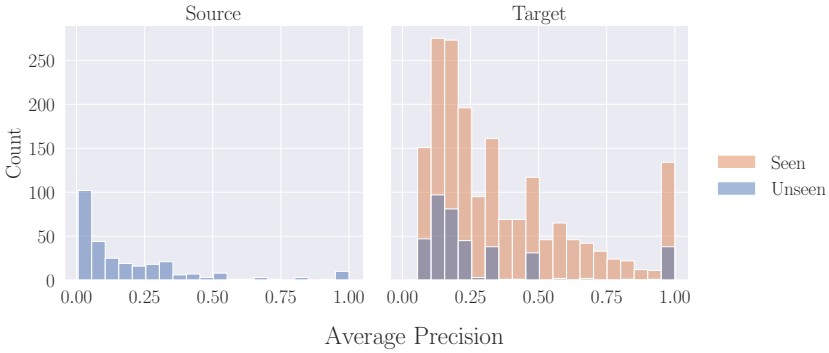

Figure 3: Average precision distributions for test set sources and targets. The GIN$_{CP}$ model was trained on cold-source split data. Each point in the histogram is a source or target node, and the color indicates whether that node had been seen during model training.

### 4.4 Zero-shot prediction potential

To assess the model's zero-shot prediction potential, where neither the source nor the target nodes are seen in training or validation, we computed the average precision (AP) scores for each source and target node to evaluate how well the model was able to retrieve its true links. In this cold-source split scenario, all source nodes in the test set are completely unseen during training, and most but not all target nodes are seen (a target node may not be seen if it is only connected to left out sources). Thus, the model's prediction of an edge with isolated endpoints, i.e. both source and target are unseen during training, is a zero-shot link prediction. As shown in Figure 3, the presence of unseen target nodes with high AP scores (see the rightmost blue bar in the Target panel) shows that for some node pairs, the model is able to predict many true links even though both the source and target nodes are isolated at test time. This suggests the model, trained on CP features and known drug-gene relationships, can predict links between completely novel drugs and genes. Further work should be done to explore the efficacy of DTI predictions in the zero-shot setting.

## 5 Discussion

Predicting the complex relationships between chemical compounds and genes is ambitious. Developing models to accurately predict these high-order interactions, particularly in inductive or zero-shot settings, is even more challenging. The MOTI$\mathcal{VE}$ dataset addresses these challenges by integrating morphological features of cells and known compound-gene relationships. By providing a large-scale, feature-rich, and extensively annotated dataset, MOTI$\mathcal{VE}$ enables the development and benchmarking of graph-based models for DTI prediction in transductive, inductive, and zero-shot scenarios. One of the key strengths of MOTI$\mathcal{VE}$ lies in its rigor. The morphological profiles were extracted

from the JUMP CP dataset using a standardized pipeline, ensuring consistency and reproducibility. Carefully curating annotations prioritized data quality and reliability. The constructed dataset also offers stringent protection against data leakage of any kind while providing thorough and challenging forms of data splitting, loading, minibatch training (requiring subgraphs), sampling, and evaluation procedures.

We acknowledge certain limitations in MOTI$\mathcal{VE}$. As discussed in Section 4.3, the effectiveness of CP features may be reduced when the graph is densely connected. Additionally, some perturbations may not induce significant morphological changes, potentially leading to uninformative node representations and false positive predictions. To address these issues, we propose isolating samples with distinguishable morphologies and reanalyzing them to assess the impact on performance. Finally, we recognize that the availability of image-based profiles remains a limitation for expanding our DTI graph; we propose that MOTI$\mathcal{VE}$ could be extended by training a generative model that translates the compound structure to in-silico Cell Painting readouts, as similarly explored in Zapata et al. [35]. Also, if the network is extended to include multiple modalities, then it could also be adapted to make predictions for nodes with missing modalities [36].

Future work may incorporate both the ORF and CRISPR gene features as a form of multimodal inputs in the graph. MOTI$\mathcal{VE}$ could also be expanded to include alternative representations for compound and gene nodes (e.g. protein structures in addition to image-based profiles) to capture known structural and sequencing relationships. The complementary information between profiles may then lead to higher quality DTI predictions. Additionally, methods may be extended to predict heterogeneous interactions rather than just binary classifications. Finally, developing end-to-end architectures to learn node embeddings directly from images could better exploit the morphological information. MOTI$\mathcal{VE}$ represents a valuable resource for the machine learning community, particularly for those interested in graph-based methods and their applications in drug discovery. By fostering interdisciplinary collaborations across graph ML, biological imaging, and drug discovery, MOTI$\mathcal{VE}$ has the potential to accelerate progress for the challenging and complex task of DTI prediction.

## Acknowledgments

The authors gratefully acknowledge an internship from the Massachusetts Life Sciences Center (to ES). We appreciate funding from the National Institutes of Health (NIH MIRA R35 GM122547 to AEC) and AEC is a Merkin Institute Fellow at the Broad Institute of MIT and Harvard.

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

# Appendix

## A  GraphSAGE$_{CP}$ forward pass algorithm

---

**Algorithm 2:** G$_{CP}$ link prediction algorithm

---

**Input:**  $G(\mathcal{V}, \mathcal{E})$

   Input features $\{\mathbf{x}_s, \forall s \in \mathcal{S}\}$ and $\{\mathbf{x}_t, \forall t \in \mathcal{T}\}$

   Embedding weight matrices $\mathbf{E}_s$ and $\mathbf{E}_t$

   Message Passing Network $\mathbf{C}_1$ and $\mathbf{C}_2$

   Non-linearity functions reLU and $\sigma$

   Neighborhood function $\mathcal{N} : v \rightarrow 2^{\mathcal{V}}$

**Output:** $\mathbf{y}_{uv}, \forall (u, v) \in$ supervision edge set $ST_{sup}$

$\mathbf{h}_s^0 \leftarrow \mathbf{x}_s \mathbf{E}_s, \forall s \in \mathcal{S}$                  ▷ Map $X_s$ to shared ft. space;

$\mathbf{h}_t^0 \leftarrow \mathbf{x}_t \mathbf{E}_t, \forall t \in \mathcal{T}$                  ▷ Map $X_t$ to shared ft. space;

**for** $v \in \mathcal{V}$ **do**

   $\mathbf{h}_v^1 \leftarrow \mathbf{C}_1(\{h_i^0; \forall i \in \mathcal{N}(v)\})$;

   $\mathbf{h}_v^1 \leftarrow \text{reLU}(\frac{\mathbf{h}_v^1}{\|\mathbf{h}_v^1\|_2})$;

**end**

**for** $v \in \mathcal{V}$ **do**

   $\mathbf{h}_v^2 \leftarrow \mathbf{C}_2(\{h_i^1; \forall i \in \mathcal{N}(v)\})$;

   $\mathbf{h}_v^2 \leftarrow \frac{\mathbf{h}_v^2}{\|\mathbf{h}_v^2\|_2}$;

**end**

$\mathbf{z}_v \leftarrow \mathbf{h}_v^1 + \mathbf{h}_v^2, \forall v \in \mathcal{V}$;

**for** $(u, v) \in ST_{sup}$ **do**

   $\mathbf{y}_{uv} = \sigma(\mathbf{z}_u \cdot \mathbf{z}_v)$;

**end**

---

## B  Experimental details

We performed a random hyperparameter search [37] for each model and data split for the number of hidden channels $(64, 128, 256)$, learning rate $([10^{-6}, 10^{-2}])$, and weight decay $([10^{-5}, 1])$. We sampled negative edges at a ratio of 1:1 at training and validation time, and 1:10 at test time. We trained each model for 1000 epochs, computed the Binary Cross Entropy loss between the prediction scores for the supervision edges and their ground truth labels, and used an Adam Optimizer to make weight updates. At validation and test time, we computed the F1 scores of the predicted edges using the best threshold found during validation time. We also computed two rank-based metrics, Hits@500 and Precision@500, to better capture how the model distinguishes between positive and negative samples. Hits@$k$ quantifies the fraction of positive test edges (total $= n$) that rank $(r)$ within the top $k$ negative test edge scores [3]. Precision@$k$ quantifies the fraction of the top $k$ predicted scores that are assigned to true positive edges. In both cases, $k = 500$ was chosen to isolate the around 1% of the test edge scores. To define these metrics more formally, let $k^-$ represent the rank of the $k$th-ranked negative edge. Then, the following equations apply:

$$H_k(r_1, r_2, \ldots, r_n) = \frac{1}{n} \Sigma_{i=1}^n \mathbb{I}[r_i \leq k^-], \quad \mathbb{I} = \begin{cases} 1 & r_i \leq k^- \\ 0 & r_i > k^- \end{cases} \tag{1}$$

$$P_k(r_1, r_2, \ldots, r_k) = \frac{1}{k} \Sigma_{i=1}^k \mathbb{I}[r_i], \quad \mathbb{I} = \begin{cases} 1 & i \text{ is a positive edge} \\ 0 & i \text{ is a negative edge} \end{cases} \tag{2}$$

We ran each scenario five times, and we reported the average performance for each model as well as the standard deviation scores in Tables 3 and 4. We set a random seed for all runs and established the

---

[3] https://ogb.stanford.edu/docs/linkprop/

same data split for each of our splitting methods, such that all variance in performance must come from batch sampling by the data loader and GPU non-determinism. All experiments ran in a NixOS server with a AMD Ryzen Threadripper PRO 7995WX processor and an NVIDIA RTX 6000 GPU card.

