# DATASHEET:
# MOTI$\mathcal{VE}$

John Arevalo    Ellen Su    Anne E. Carpenter    Shantanu Singh
Imaging Platform - Broad Institute of MIT and Harvard
{jarevalo, suellen, anne, shantanu}@broadinstitute.org

**This document is based on *Datasheets for Datasets* by Gebru *et al.* [1]. Please see the most updated version [here](here).**

## MOTIVATION

**For what purpose was the dataset created?** Was there a specific task in mind? Was there a specific gap that needed to be filled? Please provide a description.
The MOTI$\mathcal{VE}$ dataset was created to promote the development of new drug-target interaction (DTI) prediction models based on both, existing relationships between compounds and their protein targets, and the similarity of JUMP Cell Painting morphological features of perturbed cells [2].The MOTI$\mathcal{VE}$ dataset was created with the DTI task in mind, and addresses a lack of graph-based biological datasets with empirical node features.

**Who created this dataset (e.g., which team, research group) and on behalf of which entity (e.g., company, institution, organization)?**
This dataset was created by the Carpenter-Singh Lab in the Imaging Platform at the Broad Institute of MIT and Harvard, Cambridge, Massachusetts.

**What support was needed to make this dataset?** (e.g.who funded the creation of the dataset? If there is an associated grant, provide the name of the grantor and the grant name and number, or if it was supported by a company or government agency, give those details.)

The authors gratefully acknowledge an internship from the Massachusetts Life Sciences Center (to ES). We appreciate funding from the National Institutes of Health (NIH MIRA R35 GM122547 to AEC) and AEC is a Merkin Institute Fellow at the Broad Institute of MIT and Harvard.

**Any other comments?**
None.

## COMPOSITION

**What do the instances that comprise the dataset represent (e.g., documents, photos, people, countries)?** Are there multiple types of instances (e.g., movies, users, and ratings; people and interactions between them; nodes and edges)? Please provide a description.
The instances of this graph-based dataset comprise compounds and genes (as two types of nodes). The edge labels represent the interaction (binding, inhibition, activation, etc) between the nodes. Link prediction on this dataset is a multi-instance prediction task [3].

**How many instances are there in total (of each type, if appropriate)?**
In the full version of MOTI$\mathcal{VE}$, there are 3,632 compound nodes, 11,509 gene nodes, and 303,156 edges consisting of 24,798 compound-gene edges, 75,330 compound-compound edges, and 203,028 gene-gene edges.

**Does the dataset contain all possible instances or is it a sample (not necessarily random) of instances from a larger set?** If the dataset is a sample, then what is the larger set? Is the sample representative of the larger set (e.g., geographic coverage)? If so, please describe how this representativeness was validated/verified. If it is not representative of the larger set, please describe why not (e.g., to cover a more diverse range of instances, because instances were withheld or unavailable).
The dataset is a sample of instances, created by merging a collection of compound and gene interactions from seven publicly available databases (BioKG [4], DGIdb[5], DRHub [6], Hetionet [7], OpenBioLink [8], PharMeBINet [9], and PrimeKG [10]) and the compounds and gene pertubations with available features from Cell Painting. The larger dataset would be all known compounds and genes and all known relationships between them. No explicit tests of representativeness were conducted, but the compound and genes included in the Cell Painting dataset were curated by the JUMP Consortium. In particular, the CRISPR genes were selected by disease-relevancy and the ORF genes were selected to represent small proteins which express well in cells. The compounds were independently nominated by each institution in the consortium. In addition, the public interaction databases are likely to include annotations for commonly studied compounds and genes. Thus, we have strong reason to believe the compounds and genes included in MOTI$\mathcal{VE}$ are highly relevant entities.

**What data does each instance consist of?** "Raw" data (e.g., unprocessed text or images) or features? In either

case, please provide a description.

Each instance (compound or gene node) is represented by a feature vector that captures a cell's morphological profile after being perturbed by that compound or genetic modification. The features were produced by using CellProfiler software to segment single cells from the Cell Painting images, batch correcting, preprocessing, and aggregating according to the procedures set out in [11]. The resulting compound node feature vectors are 737-dimensional, and the gene node feature vectors are 722-dimensional. The edges are represented by a tuple pair of node ids, and indicate a relationship between the two nodes. The DTI prediction task is a multi-instance prediction task [3].

**Is there a label or target associated with each instance?** If so, please provide a description.

There is no label associated with each node. We do provide metadata mappings between the compound node ids and their InChIKey identifiers and the gene node ids and their Gene Symbols. The edges do have binary labels which indicate if two instances interact, and the edge metadata tables record the interaction types, e.g., binding, inhibition, activation.

**Is any information missing from individual instances?** If so, please provide a description, explaining why this information is missing (e.g., because it was unavailable). This does not include intentionally removed information, but might include, e.g., redacted text.

There is no information missing from any instance but please note that the dataset does not contain all possible instances (see above).

**Are relationships between individual instances made explicit (e.g., users' movie ratings, social network links)?** If so, please describe how these relationships are made explicit.

The edge labels represent the relationship between the nodes. The labels are binarized in the graph dataset (relationship exists/does not exist), but the original relationship labels are preserved and provided as metadata in the raw edges files of all three edge sets.

**Are there recommended data splits (e.g., training, development/validation, testing)?** If so, please provide a description of these splits, explaining the rationale behind them.

The recommended data splits are random splitting or cold start splitting, by compounds or by genes. Random splitting partitions all of the compound-gene edges into train, validation, and test sets in a 70/10/20 ratio, and keeps all of the compound-compound and gene-gene edges in the training data. Cold compound splitting partitions the compound nodes individually into train, validation, and test sets in a 70/10/20 ratio, then partitions all of the subsequent edges according to the label of the compound. Cold gene

splitting is implemented symmetrically, and splits the edges by the partitioned gene nodes. These latter two splits are needed for inductive link prediction on new, unseen compounds or genes.

**Are there any errors, sources of noise, or redundancies in the dataset?** If so, please provide a description.

We are not aware of systematic errors in the dataset; by nature, the databases containing information about genes and compounds is incomplete and subject to various technical noise.

**Is the dataset self-contained, or does it link to or otherwise rely on external resources (e.g., websites, tweets, other datasets)?** If it links to or relies on external resources, a) are there guarantees that they will exist, and remain constant, over time; b) are there official archival versions of the complete dataset (i.e., including the external resources as they existed at the time the dataset was created); c) are there any restrictions (e.g., licenses, fees) associated with any of the external resources that might apply to a future user? Please provide descriptions of all external resources and any restrictions associated with them, as well as links or other access points, as appropriate.

This dataset is self-contained. It comprises the JUMP Cell Painting data and the 7 knowledge graphs and databases previously discussed. All of its dependencies are from publicly available databases. As the interaction databases may grow in the future, the pipeline to merge these resources may create larger MOTI$\mathcal{VE}$ graphs. At the time of publication, we have frozen and released the full sets of edges and node profiles that exist in the merged graph, such that our existing version of MOTI$\mathcal{VE}$ may always be reproduced.

**Does the dataset contain data that might be considered confidential (e.g., data that is protected by legal privilege or by doctor-patient confidentiality, data that includes the content of individuals' non-public communications)?** If so, please provide a description.
None.

**Does the dataset contain data that, if viewed directly, might be offensive, insulting, threatening, or might otherwise cause anxiety?** If so, please describe why.
None.

**Does the dataset relate to people?** If not, you may skip the remaining questions in this section.

No. The cellular images used to generate the node features are all taken in vitro. The cell line is a commonly-used historical line derived from a white female patient. Therefore, conclusions from this data may only hold true for the demographics or genomics of those persons and not broader groups. U2OS was chosen because it is well-suited for microscopy, and it offers the advantage of enabling direct comparison to extensive prior studies using them.

**Does the dataset identify any subpopulations (e.g., by age, gender)?** If so, please describe how these subpopulations are identified and provide a description of their respective distributions within the dataset.
N/A.

**Is it possible to identify individuals (i.e., one or more natural persons), either directly or indirectly (i.e., in combination with other data) from the dataset?** If so, please describe how.
N/A.

**Does the dataset contain data that might be considered sensitive in any way (e.g., data that reveals racial or ethnic origins, sexual orientations, religious beliefs, political opinions or union memberships, or locations; financial or health data; biometric or genetic data; forms of government identification, such as social security numbers; criminal history)?** If so, please provide a description.
N/A.

**Any other comments?**
None.

---

## COLLECTION

**How was the data associated with each instance acquired?** Was the data directly observable (e.g., raw text, movie ratings), reported by subjects (e.g., survey responses), or indirectly inferred/derived from other data (e.g., part-of-speech tags, model-based guesses for age or language)? If data was reported by subjects or indirectly inferred/derived from other data, was the data validated/verified? If so, please describe how.
The Cell Painting data associated with each instance was directly observable as image feature vectors. The edges are inferred indirectly from other data which may be structure based similarity, known biochemical mechanisms of action, etc.

**Over what timeframe was the data collected?** Does this timeframe match the creation timeframe of the data associated with the instances (e.g., recent crawl of old news articles)? If not, please describe the timeframe in which the data associated with the instances was created. Finally, list when the dataset was first published.
The JUMP Cell Painting Consortium ran the perturbation experiments between 2020 and 2022. Annotations were collected from datasets published between 2017 and 2023.

**What mechanisms or procedures were used to collect the data (e.g., hardware apparatus or sensor, manual human curation, software program, software API)?**

How were these mechanisms or procedures validated?
For the node features, CellProfiler software was used to segment and profile each cellular image, before manual correcting and preprocessing. For the edges, manual curation was required to select the interaction types of interest.

**What was the resource cost of collecting the data?** (e.g. what were the required computational resources, and the associated financial costs, and energy consumption - estimate the carbon footprint. See Strubell *et al.*[**?**] for approaches in this area.)
There was negligible cost to collect the data for MOTI$\mathcal{VE}$. All of the primary databases are public, and the aggregation of the interactions and profiles took minimal computational resources. Thus, the creation of MOTI$\mathcal{VE}$ itself did not take energy costs.

**If the dataset is a sample from a larger set, what was the sampling strategy (e.g., deterministic, probabilistic with specific sampling probabilities)?**
We did not use any sampling procedures during dataset collection.

**Who was involved in the data collection process (e.g., students, crowdworkers, contractors) and how were they compensated (e.g., how much were crowdworkers paid)?**

The authors of this paper, all members of the Carpenter-Singh lab, are soley responsible for collecting this data. All funding sources are listed in the Motivation section.

**Were any ethical review processes conducted (e.g., by an institutional review board)?** If so, please provide a description of these review processes, including the outcomes, as well as a link or other access point to any supporting documentation.
None.

**Does the dataset relate to people?** If not, you may skip the remainder of the questions in this section.

No. The cellular images used to generate the node features are all taken in vitro. The cell line is a commonly-used historical line derived from a white female patient. Therefore, conclusions from this data may only hold true for the demographics or genomics of those persons and not broader groups. U2OS was chosen because it is well-suited for microscopy, and it offers the advantage of enabling direct comparison to extensive prior studies using them.

**Did you collect the data from the individuals in question directly, or obtain it via third parties or other sources (e.g., websites)?**
N/A.

**Were the individuals in question notified about the data collection?** If so, please describe (or show with

screenshots or other information) how notice was provided, and provide a link or other access point to, or otherwise reproduce, the exact language of the notification itself.
N/A.

**Did the individuals in question consent to the collection and use of their data?** If so, please describe (or show with screenshots or other information) how consent was requested and provided, and provide a link or other access point to, or otherwise reproduce, the exact language to which the individuals consented.
N/A.

**If consent was obtained, were the consenting individuals provided with a mechanism to revoke their consent in the future or for certain uses?** If so, please provide a description, as well as a link or other access point to the mechanism (if appropriate)
N/A.

**Has an analysis of the potential impact of the dataset and its use on data subjects (e.g., a data protection impact analysis)been conducted?** If so, please provide a description of this analysis, including the outcomes, as well as a link or other access point to any supporting documentation.
N/A.

**Any other comments?**
None.

---

### PREPROCESSING / CLEANING / LABELING

**Was any preprocessing/cleaning/labeling of the data done(e.g.,discretization or bucketing, tokenization, part-of-speech tagging, SIFT feature extraction, removal of instances, processing of missing values)?** If so, please provide a description. If not, you may skip the remainder of the questions in this section.
The raw features were preprocessed by existing protocols in order to clip outliers, correct batch effects, and aggregate over cells and replicates. The metadata labels for each compound node were also shortened to the first 14 characters of the InChIKeys such that the merge with the Cell Painting compounds was more generous. The edge relationships were preprocessed by dropping duplicate and self loop edges, as the edge set aggregated many databases with shared sources. Finally, during the merge with the JUMP Cell Painting data, the nodes in each set of edge types (compound-compound, compound-gene, and gene-gene) were pruned out such that the maximum node degree of each entity was 150 for that type. This pruning was done iteratively with an algorithm inspired by [12], and was necessary so that no instance dominated the graph structure.

**Was the "raw" data saved in addition to the preprocessed/cleaned/labeled data (e.g., to support unanticipated future uses)?** If so, please provide a link or other access point to the "raw" data.
Yes. Although the raw data still exists in the numerous databases used to compile MOTI$\mathcal{VE}$, we have saved all the input versions used to compile MOTI$\mathcal{VE}$. This unprocessed and unmerged data is provided should a user choose to run or adapt the processing pipeline independently. More details on the input data and its access can be found in the GitHub repository wiki page.

**Is the software used to preprocess/clean/label the instances available?** If so, please provide a link or other access point.
Yes, the code used to prepare the data is available at `https://github.com/carpenter-singh-lab/motive`.

**Any other comments?**
None.

---

### USES

**Has the dataset been used for any tasks already?** If so, please provide a description.
The dataset, as assembled, has only been used to make DTI predictions in the MOTI$\mathcal{VE}$ paper. The underlying data (JUMP Cell Painting and the various other databases) has been used in many other research projects.

**Is there a repository that links to any or all papers or systems that use the dataset?** If so, please provide a link or other access point.
There is none at the moment.

**What (other) tasks could the dataset be used for?**
MOTI$\mathcal{VE}$ can be used to develop DTI prediction models. Or, if added as an additional modality when building biological knowledge graphs, it may be used in any downstream prediction tasks associated with any biological graphs.

**Is there anything about the composition of the dataset or the way it was collected and preprocessed/cleaned/labeled that might impact future uses?** For example, is there anything that a future user might need to know to avoid uses that could result in unfair treatment of individuals or groups (e.g., stereotyping, quality of service issues) or other undesirable harms (e.g., financial harms, legal risks) If so, please provide a description. Is there anything a future user could do to mitigate these undesirable harms?
No. The raw data for MOTI$\mathcal{VE}$ already existed publicly, and there is no information in this dataset that corresponds to individuals or groups.

**Are there tasks for which the dataset should not be**

**used?** If so, please provide a description.
Although this dataset should be used to make DTI predictions, the predicted therapeutics should still be tested before clinical use. It should not be used to design harmful chemicals.

**Any other comments?**
None.

## DISTRIBUTION

**Will the dataset be distributed to third parties outside of the entity (e.g., company, institution, organization) on behalf of which the dataset was created?** If so, please provide a description.
Yes, the dataset is publicly available on the internet.

**How will the dataset will be distributed (e.g., tarball on website, API, GitHub)?** Does the dataset have a digital object identifier (DOI)?
The processed dataset is available at `https://cellpainting-gallery.s3.amazonaws.com/index.html#cpg0034-arevalo-su-motive/broad/workspace/publication_data/2024_MOTIVE`. The code to process the graph data from the raw files is available at `https://github.com/carpenter-singh-lab/motive`.

**When will the dataset be distributed?**
The dataset was first released in June 2024.

**Will the dataset be distributed under a copyright or other intellectual property (IP) license, and/or under applicable terms of use (ToU)?** If so, please describe this license and/or ToU, and provide a link or other access point to, or otherwise reproduce, any relevant licensing terms or ToU, as well as any fees associated with these restrictions.
We release the source code to process the collected data under a BSD 3-Clause License. JUMP Cell painting data is released with CC0 1.0 Universal (CC0 1.0). We also aggregate content from seven external databases with varying content licenses. The copyright of the data belongs to the authors of the original databases.

**Have any third parties imposed IP-based or other restrictions on the data associated with the instances?** If so, please describe these restrictions, and provide a link or other access point to, or otherwise reproduce, any relevant licensing terms, as well as any fees associated with these restrictions.
Yes, see answer to prior question.

**Do any export controls or other regulatory restrictions apply to the dataset or to individual instances?** If so, please describe these restrictions, and provide a link or other access point to, or otherwise reproduce, any supporting

documentation.
None.

**Any other comments?**
None.

## MAINTENANCE

**Who is supporting/hosting/maintaining the dataset?**
The Carpenter-Singh Lab at Broad Institute of MIT and Harvard is supporting and maintaining the dataset.

**How can the owner/curator/manager of the dataset be contacted (e.g., email address)?**
The Carpenter-Singh Lab at Broad Institute of MIT and Harvard can be contacted at `imagingadmin@broadinstitute.org` or `https://carpenter-singh-lab.broadinstitute.org/more_info`.

**Is there an erratum?** If so, please provide a link or other access point.
There is no explicit erratum, but the log of updates to the data preparation pipeline and preprocessing steps and previously known errors can be found in the GitHub repository.

**Will the dataset be updated (e.g., to correct labeling errors, add new instances, delete instances)?** If so, please describe how often, by whom, and how updates will be communicated to users (e.g., mailing list, GitHub)?
If there are any updates to the dataset coming from the discovery and publication of additional compound and gene interactions, new versions will be published at the same location. We expect the dataset to be a fairly static resource, but if there are large changes we will also publish a notification in the GitHub repository.

**If the dataset relates to people, are there applicable limits on the retention of the data associated with the instances (e.g., were individuals in question told that their data would be retained for a fixed period of time and then deleted)?** If so, please describe these limits and explain how they will be enforced.
N/A.

**Will older versions of the dataset continue to be supported/hosted/maintained?** If so, please describe how. If not, please describe how its obsolescence will be communicated to users.
The older versions of the dataset will be kept for reproducibility of previous works.

**If others want to extend/augment/build on/contribute to the dataset, is there a mechanism for them to do so?** If so, please provide a description. Will these contributions

be validated/verified? If so, please describe how. If not, why not? Is there a process for communicating/distributing these contributions to other users? If so, please provide a description.

We welcome contributions to the dataset. We published this resource as part of the Cell Painting Gallery data repository, which has established a guideline for contribution, validation and distribution at `https://broadinstitute.github.io/cellpainting-gallery/contributing_to_cpg`. We also welcome contributions and feedback in the GitHub repository `https://github.com/carpenter-singh-lab/motive`

**Any other comments?**

None.