# OpenReview forum: "MOTIVE: A Drug-Target Interaction Graph For Inductive Link Prediction"
_NeurIPS.cc/2024/Datasets_and_Benchmarks_Track — NeurIPS 2024 Track Datasets and Benchmarks Spotlight_

### Official Review · Reviewer_AitS · 2024-07-08
**Review for Paper 1374**

**Rating:** 6
**Confidence:** 5
**Correctness:** yes
**Clarity:** yes

**Review:**

The paper constructed a dataset from the public available datasets where Cell Painting features were used for the task of DTI prediction. Although there existed several contributions in the paper, it still had limitations. The advantages and disadvantages of the paper were listed below.

Strong points:
1) Authors analyzed the limitations of existing DTI datasets, such as the lack of empirical node features,  and the limited volume of existing datasets.
2) The paper constructed a DTI datasets extracted from the seven public available compound and gene relevant dataset and used the Cell Painting features as the initial features.
3) This paper designed the data splitting strategy and the negative sampling method in the case of no data leaking under the setting of the inductive learning and run the inductive learning model to prove it.

Weak points:
1) Although this paper proposed a DTI dataset, the contributions of this paper are very limited. Firstly, the value of the dataset is limited since it is can be only used for link prediction. And there is no challenge for data construction and preprocessing, such as data collection and data cleaning. Those lead to the limited contribution of this paper.
2) The paper mentioned that the model with Cell Painting features achieved the best performance compared to graph structure alone, feature-based models, and topological heuristics, however, there are no relevant experimental results to support all these points. In addition, the paper lacks the exploration of other features, such as word2vec. With the advent of the large language models, some of them can be utilized for node feature initialization.
3) The paper only utilized the simple existing models to run experiments. Some SOTA methods should be used to prove the validness of the constructed datasets. This greatly hinders the confidence of the dataset. In addition, the used metrics are inadequate as well as the setting of K in hit ratio and precision.
4) In the data splitting, although GNN models can update node embeddings from neighbors by message passing, how the isolated nodes benefit from GNNs? How the node embeddings are updated for them? More detail should be provided.
5) In the related work Section, the paper mentioned that existing methods either cannot capture higher-order relationships, or did not exploit the network connectivity across compounds and genes. How these problems are solved by this paper? A detailed discuss should be offered.

**Strengths:**

Please refer to Review Section.

**Additional Feedback:**

Please refer to Review Section.

**Documentation:**

no

**Limitations:**

Please refer to Review Section.

**Opportunities For Improvement:**

Please refer to Review Section.

**Relation To Prior Work:**

yes

**Summary And Contributions:**

This paper focused on the problem of drug-target interaction (DTI) prediction. To promote the DTI, authors contributed a large-scale dataset using the Cell Painting features extracted from the seven public available datasets relevant to compound and genes. In addition, the paper provided the dataset splitting strategy for cold-start setting under the inductive learning framework as well as the method for negative data sampling.

---

> ### Author Rebuttal · Authors · 2024-08-15
>
> ## 1. Contribution and generalizability
>
> We thank the reviewer for the helpful comments and suggestions and acknowledge the reviewer's concerns. We believe the DTI task itself is a major challenge with strong practical relevance for drug discovery (both for identifying new chemicals that target a protein ("Hit Identification", and for identifying the protein target of a chemical of interest ("Target Deconvolution" - these are two major steps in preclinical research) Developing more powerful link prediction methods is therefore itself extremely useful. We will add text explaining this in the Intro.
>
> As well, the final representations learned by the GNNs can be used in any downstream graph ML tasks (MoA detection, node or graph classification, etc) beyond DTI. Further, the dataset construction and cleaning process was nontrivial and required a thorough understanding of the shared public databases; we regret leaving the impression it was not a crucial part of the benchmark we present. We extended the description of the morphological profiles preprocessing as follows:
>
> > Data preparation for morphological profiles should be finetuned depending on the assay conditions. For MOTIVE, we applied the protocols from Arevalo et al. [21] and Chandrasekaran et al. [6], extensively optimizing the pipelines separately for compound and genetic perturbations. For compound perturbations, we first filtered out features with low variance, then applied median absolute deviation normalization, a rank-inverse normal transformation, and Harmony [[korsunsky2019fast](https://www.nature.com/articles/s41592-019-0619-0)] to reduce the batch effects. Features were selected based on correlation analysis. For genetic perturbations, the optimal pipeline differed slightly: we subtracted the mean vector per well from each feature vector to account for well position effects and replaced the INT transformation with an outlier removal step. Finally, we aggregated the replicates (usually five) of each perturbation using median profiles to make the representations robust to low-quality images. In our experience, these comprised less than 5% of the data and were uncorrelated across replicates on different plates.
>
> ## 2. Baselines and alternative representations
>
> Table 3, section 4.1, shows the benchmarking results the reviewer was looking for; we welcome guidance on how to improve the data presentation that made this hard to find or confusing.
>
> In regards to using word2vec or LLMs for node feature initialization, we wonder whether the reviewer suggests using textual embeddings of the gene names or other text associated with the node entities, e.g. SMILEs strings for compounds. As discussed above, we find alternative node representations to be an interesting and worthwhile avenue of improvement, and we are open to this discussion as part of our future work. We do note that such representations are rarely able to represent both chemical and gene perturbations, which is a main strength of the image-based representations. We plan to clarify this in our paper as:
>
> > Using image-based profiles also offers the advantage of representing both compound and gene nodes with the same modality (as opposed to SMILE based or protein structure based), ensuring similar statistical properties and making it easier to discover cross-modal relationships [[srivastava2014multimodal](https://jmlr.org/papers/volume15/srivastava14b/srivastava14b.pdf)].
>
> ## 3. SOTA models and metrics
>
> We took the reviewer's suggestion and tested the GIN [ICLR 2019] and GATv2 [ICLR 2022] architectures. Although there is not a consensus on the best GNN architecture, the superiority of combining CellPainting information with Graph-based methods remains for the cold-source scenario and is larger for the cold-target scenario.
>
> | Split   | Model   | F1            | Hits@500      | Precision@500   |
> |:--------|:--------|:--------------|:--------------|:----------------|
> | Source  | GIN     | 0.2888±0.0544 | 0.1278±0.0096 | 0.6240±0.0278   |
> |         | GATv2   | 0.3189±0.0229 | 0.0606±0.0068 | 0.3608±0.0230   |
> | Target  | GIN     | 0.3665±0.1309 | 0.2574±0.0197 | 0.8468±0.0268   |
> |         | GATv2   | 0.1699±0.0073 | 0.0963±0.0156 | 0.5020±0.0867   |
>
>
> We would like to understand better the reviewer’s concern about the metrics we used. Hits@k and precision@k are both widely used metrics in the link prediction and graph deep learning community. Notably, the Open Graph Benchmark library uses these metrics to evaluate each graphML dataset. The k term is particularly relevant for DTI, as it may represent the number of compounds a company tests in followup assays, for example.
>
> ## 4. Node embeddings
>
> The reviewer is correct that the feature vectors of isolated nodes are not updated by the GNN as there are no neighbors features to aggregate. In this case (Section 4.2: Inductive link prediction), the GNN relies on available information from the Cell Painting features of isolated nodes and the learned embeddings of trained nodes to make link predictions. The fully isolated case (i.e. zero-shot learning) is discussed in Section 4.4.
>
> ## 5. Advancements in capturing higher-order relationships and exploiting network connectivity
>
> We thank the reviewer for pointing this out. We updated the related work section to better present the features of our work:
>
> > Rohban et al. matched compounds and genes based on feature vector similarities. In contrast, MOTIVE enables the usage of machine learning methods to capture nonlinear  compound-gene relationships. Next, Herman et al. makes use of chemical structures and morphological profiles to predict toxicity assays without considering the connectivity network of compound gene interactions. MOTIVE also expands this by incorporating the morphological profiles of both compounds and genes in a graph and using the message passing framework to leverage such network connectivity.

---

> > ### Comment · Reviewer_AitS · 2024-08-31
> > **Official Comment by Reviewer AitS**
> >
> > Dear Authors,
> >
> > Thanks for your detailed response. I believe you solved my concerns, therefore, I prefer to increase my score to 6. Thank you again.
> >
> > Best regards,
> >
> >  Reviewer AitS

---

> > > ### Author Response · Authors · 2024-09-04
> > >
> > > Dear reviewer,
> > >
> > > Thanks for your feedback and for deciding to increase the score. We noticed the score hasn't been updated yet and wanted to check if everything went through correctly.
> > >
> > > We appreciate your time and attention!

---

### Official Review · Reviewer_gY9F · 2024-07-21
**Dataset combining drug discovery knowledge graph with Cell Painting**

**Rating:** 8
**Confidence:** 4
**Correctness:** N/A
**Clarity:** N/A

**Review:**

Overall a clearly written, high-quality paper that addresses a significant gap in current datasets and demonstrates that combining ground-truth knowledge graph with imaging data can boost performance.

**Strengths:**

- Previously unexplored dataset niche.
- Significant attention given to various split scenarios, which translates to multiple potential drug discovery tasks being covered.
- Reasonably extensive evaluation, showcasing made claims.

**Additional Feedback:**

N/A

**Documentation:**

N/A

**Limitations:**

As discussed, could benefit from discussion on how to approach lack of image data for new compounds.

**Opportunities For Improvement:**

- In the current form, the proposed approach relies on CP features for prediction on novel drugs. This can be very liming, since the number of publicly available screened compounds, even in JUMP, is still relatively small. A discussion on how to mitigate this issue would be beneficial, for instance I wonder if the authors tested simply using embeddings computed based on molecular structures instead. In general, I'd say that the lack of such experiment is the biggest shortcoming of the paper (the paper drives the point that *some* node representation is needed, but not necessarily that it needs to be the *image* representation).
- Some formatting issues, e.g. with section capitalization.

**Relation To Prior Work:**

N/A

**Summary And Contributions:**

Authors combine known interactions between drugs and genes into a single knowledge graph, and cross it over with JUMP Cell Painting data. They design several splitting strategies that mimic different drug discovery scenarios, and experimentally show usefulness of using image representations in combination with the knowledge graph.

---

> ### Author Rebuttal · Authors · 2024-08-15
>
> ## Formatting
>
> We thank the reviewer for taking the time to provide helpful comments and feedback. First, to address the formatting mishap, we have made the changes so that all section titles are in lower case except for the first word and proper nouns.
>
> ## Alternative node embeddings
>
> To this point, we thank the reviewer for thinking of this more comprehensive experimentation plan. We did consider using structure based representation methods (such as Mol2Vec and protein sequencing), but, ultimately, the focus of our work is that image-based profiles contribute useful information to making DTI predictions, rather than necessarily being more informative than any other representation. We will clarify our claims in the Introduction section as follows:
>
> > Using image-based profiles also offers the advantage of representing both compound and gene nodes with the same modality (as opposed to SMILE based or protein structure based), ensuring similar statistical properties and making it easier to discover cross-modal relationships [[srivastava2014multimodal](https://jmlr.org/papers/volume15/srivastava14b/srivastava14b.pdf)].
>
> Additionally, we took extreme care to avoid any information leakage in our data collection, preprocessing, splitting, and sampling; with the addition of structural representations, we run the risk of leaking relational information between the databases used to produce the structures and our dataset. We concur that additional modalities of compound structure and gene representations would also contribute useful information for this task. Thus, we will add in our Discussion of future work:
>
> > As an open area of future research, MOTIVE may be expanded to include alternative representations of each compound and gene node to capture chemical and protein structures and sequencing data. The complementary information between profiles may lead to higher quality DTI predictions.
>
>
> ## Lack of image data for new compounds
>
> We agree with the reviewer that the fact that image-based profiles do not exist for all compounds and genes is currently limiting. However, it is noteworthy that Cell Painting is a much less expensive assay than, say, genomics, transcriptomics, and proteomics - we therefore anticipate public databases to eventually surpass those in size. Certainly in the context of a pharmaceutical company or even an academic laboratory, running the assay on newly synthesized compounds would be a relatively minimal investment. Nevertheless, the creation of models that can leverage Cell Painting information and also explore the full chemical space is an exciting research opportunity. We add some alternatives to explore in the discussion section:
>
> > One option to extend MOTIVE to compounds that do not have an image-based profile is to train a generative model that translates the compound structure to in-silico Cell Painting readouts, as similarly explored in Zapata et al. [[zapata2023cell](https://pubs.rsc.org/en/content/articlehtml/2023/dd/d2dd00081d)]. Also, If the network is extended to include multiple modalities, then it could also be adapted to make predictions for nodes with missing modalities [[wang2023multi](https://openaccess.thecvf.com/content/CVPR2023/papers/Wang_Multi-Modal_Learning_With_Missing_Modality_via_Shared-Specific_Feature_Modelling_CVPR_2023_paper.pdf)].

---

### Official Review · Reviewer_QTzg · 2024-07-24
**Review of "MOTIVE : A Drug-Target Interaction Graph For Inductive Link Prediction"**

**Rating:** 7
**Confidence:** 3

**Review:**

The article is interesting and well written. I think that the proposed graph is novel, and potentially useful for further research in the DTI domain.

Pros:

- Novel graph dataset providing DTI data with new features (Cell Painting).
- Well written, and interesting.
- Potential good impact for the DTI research commnunity.

Cons:

I do not find any major cons.

---

After reading the authors' rebuttal, I have increased my rating from 5 to 7.

**Strengths:**

I think the main strengths of this article are:
- clearly written
- potentially very interesting contribution for further research on DTI: Cell Painting data represents a new modality, that to the best of my knowledge, is not currently available in the form of structured graph.
- carefully designed data splits for reproducible experiments avoid data leakage.

**Additional Feedback:**

See comments/questions above.

**Clarity:**

The paper is clearly written. The only section that is not clear (in my opinion) is 4.4.

**Correctness:**

I think the dataset is constructed in a sound way, and the description is clear.

**Documentation:**

The dataset is clearly described, and the article contains sufficient details.

**Ethics:**

I cannot think of any ethical concern.

**Limitations:**

Yes.

**Opportunities For Improvement:**

Some comments/questions:
1. suggestion on Algorithm 1: instead of
[[
if |B_N| < cr then
  continue;
end
B_N' <- f(B_N, cr);
B' <- B_P U B_N';
return B'
]]
I think it is clearer if you write:
[[
if |B_N| >= cr then:
  B_N' <- f(B_N, cr);
  B' <- B_P U B_N';
  return B'
end

2. Algorithm 2:
- in the comment I think X_s and X_t should be x_s and x_t (lowercase)
- in input you define "Convolution weight matrices W^{1} and W{2}" (with superscript indices) but I think in the body of the algorithm you use them with subscript indices (W_{1} and W_{2})

3. Section 3.6, line 193: have you tried GraphSAGE with non-random initialization? For example, for targets you may use a model that encodes the protein sequence, and for compounds a model that encodes the SMILEs. It would be interesting to see how a GraphSAGE network initialized in this way compares to GraphSAGE_CP.

4. Section 4: Is the "Shortest Path" a useful/meaningful baseline? I do not think so.

5. Section 4.2: I am not sure why GraphSAGE_embs is not applicable: has it not the same architecture as GraphSAGE_CP? Are you training the projection layers? Why is GraphSAGE_CP applicable and GraphSAGE_embs is not?

6 Section 4.4 and Figure 3: I think this part and the figure are not clear.

]]

**Relation To Prior Work:**

Yes.

**Summary And Contributions:**

The article presents a new graph dataset to study drug-target interaction (DTI). The novel proposed graph dataset includes features for genes and compounds obtained with the Cell Painting method. The authors clearly describe the motivation and the related works. They provides a detailed description of the graph (and of the method used to build it); they describe the available data split, and corresponding methods. Finally they present some experiments using the proposed graph, and discuss the results.

---

> ### Author Rebuttal · Authors · 2024-08-15
>
> ## 1., 2. Algorithm and math notation
>
> We thank the reviewer for catching the capitalization error and suggesting improvements for the algorithm's readability. We have updated the paper to reflect these changes. In algorithm 2, we used the superscripts to indicate the SAGEConv layers 1 and 2, and the subscripts to indicate the 2 weight matrices that make up each SAGEConv layer. We have now listed the four W matrices as inputs in the algorithm header for clarity.
>
> ## 3. Non-random node initialization
>
> We appreciate the reviewer's suggestion to explore non-random initializations; these are great ideas. While exploring additional modalities, including structure-based representations, might contribute valuable information, our primary goal is to establish the effectiveness of image-based profiles for DTI prediction. Moreover, we took great care to avoid information leakage throughout our data collection, preprocessing, splitting, and sampling processes. When introducing structural representations, we run the risk of information leakage between the databases used to produce these structures. We will highlight the convenience of using image-based profiles for both genes and compounds as follows:
> > Using image-based profiles also offers the advantage of representing both compound and gene nodes with the same modality (as opposed to SMILE based or protein structure based), ensuring similar statistical properties and making it easier to discover cross-modal relationships [[srivastava2014multimodal](https://jmlr.org/papers/volume15/srivastava14b/srivastava14b.pdf)].
>
> We will also include non-random initialization as potential research venues:
>
> > As an open area of future research, MOTIVE may be expanded to include alternative representations of each compound and gene node to capture chemical and protein structures and sequencing data. The complementary information between profiles may lead to higher quality DTI predictions.
>
> ## 4. Shortest path baseline
>
> We included the shortest path baseline to set a lower bound for DTI performance. Although simple, we believe similar nodes will share similar neighbors and thus have shorter path lengths. Since our graph is sparse, this topological heuristic achieves extremely low scores, showing that learning-based methods are required to make high quality link predictions. We also included the MLP and Bilinear models as more meaningful, learning-based baselines.
>
> ## 5. GraphSAGE_embs is transductive
>
> The reviewer's interpretation is correct. Both GraphSAGE_CP and GraphSage_embs have the same architecture in the message passing and link prediction layers, but GraphSAGE_embs learn a representation for nodes using projection layers. Thus, GraphSage_embs is incompatible with cold split scenarios. In the paper, we will add to Section 4.2:
>
> > GraphSAGE_embs is not applicable in this scenario; as it learns representations for nodes using projection layers, it is limited to making predictions for nodes that it has already seen during training (i.e. transductive link predictions).
>
> ## 6. Zero-shot scenario (Section 4.4)
>
> Section 4.4 gives an anecdotal case of successfully predicting links for two nodes which it had never seen during training purely based on their Cell Painting (CP) feature vectors, a.k.a zero shot link prediction. We will elaborate on why we chose to use average precision scores:
>
> > To assess the model's zero-shot prediction potential, we computed the average precision scores for each source and target node to evaluate how well the model was able to retrieve its true links.
>
> We will clarify our main claim of this section, and its implications for the model’s generalizability and future applicability.
>
> > The fact that there are unseen targets with high average precision scores (see the rightmost blue bar in the Target panel) shows that for some node pairs, the model is able to predict many true links even though both the source and target nodes are isolated at test time. This suggests the model, trained on CP features and known drug gene relationships, is able to generalize what it has learned to predict links between completely novel drugs and genes.

---

> > ### Comment · Reviewer_QTzg · 2024-09-04
> >
> > Dear Authors,
> >
> > Thanks for your detailed response. You solved my concerns, and I have increased my rating from 5 to 7.
> >
> > Best regards

---

### Official Review · Reviewer_NgGB · 2024-07-26
**Novel Graph Dataset with biological meaning**

**Rating:** 8
**Confidence:** 4
**Correctness:** No concerns in this perspective.
**Clarity:** This paper is well written in general.

**Review:**

This submission proposes to add a novel feature, namely Cell Painting (CP) to construct graph features for DTI prediction. The reviewer acknowledges this is an important feature to incorporate and the authors generally conduct the construction soundly. The dataset is further used to evaluate graph models for DTI. The cold start setting justifies the generalization of the feature. Apart from some concerns, I give quite positive opinion on this paper.

**Strengths:**

1.	The dataset is based on Cell Painting (CP) assays, which generate morphological profiles of cells after they have been perturbed by chemicals or genetic edits. As said, this feature enhances a graph of compound and gene relations. This addition is crucial to improve graph-based methods.
2.	The research also evaluates various models, including GraphSAGE and other baseline models to ablate the feature contribution.
3.	The graph structure is well described and recorded to represent biological relations.

**Additional Feedback:**

NA

**Documentation:**

No concerns in this perspective.

**Ethics:**

No concerns in this perspective.

**Limitations:**

No concerns in this perspective.

**Opportunities For Improvement:**

1.	The signal and noise of CP assays is not thoroughly discussed. As a biological assay, normalization and curation steps are crucial for the quality. What are the key steps to ensure this?
2.	GraphSAGE is a quite classical method for graph learning? How does more advanced method perform on this dataset?

**Relation To Prior Work:**

The prior works are properly discussed.

**Summary And Contributions:**

In this submission, the authors propose a DTI prediction dataset with graph structure. Specifically, It includes morphological profiles extracted from the JUMP CP dataset, which allows for the representation of cells' appearances after chemical or genetic perturbations. Also, it provides four different graph structures for evaluation: bipartite, source-expanded, target-expanded, and source and target expanded. For model training, the dataset offers two types of data splits: random and cold start splits, which are useful for transductive and inductive link prediction tasks, respectively.

---

> ### Author Rebuttal · Authors · 2024-08-15
>
> ## Data quality
>
> We appreciate the reviewer's query regarding the importance of normalization and curation steps for ensuring data quality. We added more details on the morphological profile preprocessing:
> > Data preparation for morphological profiles should be finetuned depending on the assay conditions. For MOTIVE, we applied the protocols from Arevalo et al. [21] and Chandrasekaran et al. [6], extensively optimizing the pipelines separately for compound and genetic perturbations. For compound perturbations, we first filtered out features with low variance, then applied median absolute deviation normalization, a rank-inverse normal transformation, and Harmony [[korsunsky2019fast](https://www.nature.com/articles/s41592-019-0619-0)] to reduce the batch effects. Features were selected based on correlation analysis. For genetic perturbations, the optimal pipeline differed slightly: we subtracted the mean vector per well from each feature vector to account for well position effects and replaced the INT transformation with an outlier removal step. Finally, we aggregated the replicates (usually five) of each perturbation using median profiles to make the representations robust to low-quality images. In our experience, these comprised less than 5% of the data and were uncorrelated across replicates on different plates.
>
> ## Benchmarking SOTA methods
> Although newer architectures exist, GraphSAGE has often been shown to achieve the best overall performance in link prediction tasks when properly fine tuned and evaluated under realistic settings [[li2023evaluating](https://proceedings.neurips.cc/paper_files/paper/2023/file/0be50b4590f1c5fdf4c8feddd63c4f67-Paper-Datasets_and_Benchmarks.pdf)]. Still, for completeness, we have now benchmarked the GATv2 [ICLR 2022] and GIN [ICLR 2019] architectures. We found that GATv2 performs worse than GraphSAGE for both cold-splitting scenarios (leave-out-source and leave-out-target); this is consistent with the results reported in the GATv2 paper for link prediction tasks where attention-based architectures underperform classical GCN and GraphSAGE architectures. We found that GIN achieves higher Hits@500 scores than GraphSAGE.
>
> | Split   | Model   | F1            | Hits@500      | Precision@500   |
> |:--------|:--------|:--------------|:--------------|:----------------|
> | Source  | GIN     | 0.2888±0.0544 | 0.1278±0.0096 | 0.6240±0.0278   |
> |         | GATv2   | 0.3189±0.0229 | 0.0606±0.0068 | 0.3608±0.0230   |
> | Target  | GIN     | 0.3665±0.1309 | 0.2574±0.0197 | 0.8468±0.0268   |
> |         | GATv2   | 0.1699±0.0073 | 0.0963±0.0156 | 0.5020±0.0867   |
>
>
> We will include both of these models our paper and update the results accordingly.

---

### Decision · Program_Chairs · 2024-09-26

**Decision:**

Accept (Spotlight)

**Comment:**

The reviewers all agree this graph dataset is a valuable contribution to the drug-target interaction prediction.

The strengths noted by the reviewers include: (1) using cell painting data in the graph dataset that has not been used in the existing datasets; (2) good data splitting strategy and negative sampling to avoid data leakage; (3) an evaluation of various of drug-target interaction prediction methods on the dataset; and (4) a good design of the graph dataset.

The main weakness is that the cell painting data is not available for most drugs and targets, which may limit the usability. However, as more cell painting data is collected, the problem may be mitigated.

Overall, the dataset and benchmark are useful for the drug-target interaction prediction field.